# Lifestyle Habits and Mental Health in Light of the Two COVID-19 Pandemic Waves in Sweden, 2020

**DOI:** 10.3390/ijerph18063313

**Published:** 2021-03-23

**Authors:** Victoria Blom, Amanda Lönn, Björn Ekblom, Lena V. Kallings, Daniel Väisänen, Erik Hemmingsson, Gunnar Andersson, Peter Wallin, Andreas Stenling, Örjan Ekblom, Magnus Lindwall, Jane Salier Eriksson, Tobias Holmlund, Elin Ekblom-Bak

**Affiliations:** 1Department of Physical Activity and Health, The Swedish School of Sport and Health Sciences, 114 33 Stockholm, Sweden; victoria.blom@gih.se (V.B.); bjorne@gih.se (B.E.); lena.kallings@gih.se (L.V.K.); daniel.vaisanen@gih.se (D.V.); erik.hemmingsson@gih.se (E.H.); orjan.ekblom@gih.se (Ö.E.); magnus.lindwall@gih.se (M.L.); jane.saliereriksson@gih.se (J.S.E.); tobias.holmlund@gih.se (T.H.); elin.ekblombak@gih.se (E.E.-B.); 2Functional Area Occupational Therapy & Physiotherapy, Allied Health Professionals Function, Karolinska University Hospital, 171 76 Solna, Sweden; 3Research Department, HPI Health Profile Institute, 182 53 Danderyd, Sweden; gunnar.andersson@hpihealth.se (G.A.); peter.wallin@hpihealth.se (P.W.); 4Department of Psychology, Umeå University, 901 87 Umeå, Sweden; andreas.stenling@umu.se; 5Department of Sport Science and Physical Education, University of Agder, 4630 Kristiansand, Norway; 6Department of Psychology, University of Gothenburg, 405 30 Gothenburg, Sweden; 7Department of Neurobiology, Care Sciences and Society, Division of Physiotherapy, Karolinska Institute, 141 83 Stockholm, Sweden

**Keywords:** physical activity, sitting, alcohol, diet, smoking, SARS-CoV-2, Sweden, mental health, health anxiety, depression

## Abstract

The COVID-19 pandemic has become a public health emergency of international concern, which may have affected lifestyle habits and mental health. Based on national health profile assessments, this study investigated perceived changes of lifestyle habits in response to the COVID-19 pandemic and associations between perceived lifestyle changes and mental health in Swedish working adults. Among 5599 individuals (50% women, 46.3 years), the majority reported no change (sitting 77%, daily physical activity 71%, exercise 69%, diet 87%, alcohol 90%, and smoking 97%) due to the pandemic. Changes were more pronounced during the first wave (April–June) compared to the second (October–December). Women, individuals <60 years, those with a university degree, white-collar workers, and those with unhealthy lifestyle habits at baseline had higher odds of changing lifestyle habits compared to their counterparts. Negative changes in lifestyle habits and more time in a mentally passive state sitting at home were associated with higher odds of mental ill-health (including health anxiety regarding one’s own and relatives’ health, generalized anxiety and depression symptoms, and concerns regarding employment and economy). The results emphasize the need to support healthy lifestyle habits to strengthen the resilience in vulnerable groups of individuals to future viral pandemics and prevent health inequalities in society.

## 1. Introduction

The pandemic caused by the coronavirus disease 2019 (COVID-19) has become a global public health emergency. To stop the virus, confinement, social distancing, and even full lockdowns have been implemented. Under such circumstances, there is a risk for radical changes of lifestyle habits such as physical activity (PA), sedentary behavior, smoking, diet, and alcohol consumption, which have all been previously linked to morbidity and pre-mortality [1,2,3,4]. For example, both short and long bouts of regular PA have been shown to improve physical and mental health in both children and adults [1,5,6].

During the first wave of the pandemic, several lifestyle habits seem to have changed, but with mixed reports from different countries. For example, studies from Belgium, France, and Switzerland have reported a general increase in both exercise frequency and sedentary behavior [7,8]. Conversely, in Italy, total PA decreased significantly during the first COVID-19 wave as compared to before, in all age groups and especially in men [9]. Moreover, several studies have shown small changes in dietary habits [10,11,12], while others have reported an increase in unhealthy food intake, overeating, and snacking between meals [10,13,14,15]. Similarly, studies have indicated that alcohol consumption has not changed during home confinement [13,16], while others have reported increased alcohol consumption [15,17,18]. Smoking has been reported to both have increased [17,19] and decreased [16,20] during the first wave of COVID-19.

Negative changes in lifestyle habits and an increased risk of depression, anxiety, and stress symptoms during the COVID-19 pandemic have been reported [17,21], while a positive association between more time spent in moderate-to-vigorous PA and better mental health has also been found [9,22,23]. However, previous studies have investigated changes in lifestyle during COVID-19 in a relatively short timeframe during the spring and summer of 2020. As the pandemic has continued, we need to examine longer-term effects on lifestyle and mental health, including comparing differences between the different waves of the pandemic. Also, with different governments employing varying countermeasures and social restrictions, it is important to study the effects on lifestyle habits and health experiences in the context of different countries. Sweden is one of the countries that has caught attention worldwide as the government chose to implement mainly recommended restrictions without any full-scale lockdown. Any comparative results from such a strategy on lifestyle habits and mental health is highly relevant for future decision making in similar situations.

The main aim of the present study was therefore to investigate perceived changes in time spent sitting, daily PA, exercise, diet, alcohol, and smoking in response to the COVID-19 pandemic in Swedish working adults, and to study potential differences across age, sex, education, occupational groups, and different waves of the pandemic. An additional aim was to study the odds ratio of perceived mental ill-health in relation to perceived lifestyle changes.

## 2. Materials and Methods

### 2.1. Study Population

Data originated from the Health Profile Assessment (HPA) database (http://www.hpihealth.se (assessed on 3 December 2020)) which contains data from HPAs carried out in health services all around Sweden since the middle of the 1970s. An HPA includes a questionnaire about lifestyle and health experiences, measurements of anthropometrics and blood pressure, estimations of maximal oxygen consumption from a submaximal cycle ergometer test, and a person-centered dialogue with an HPA coach. An HPA is offered to all employees working for a company or an organization connected to occupational or health-related services, and is voluntary and free of charge for the employee. All data are subsequently recorded in the Health Profile Institute database. In the light of the COVID-19 pandemic emerging in March 2020, additional questions regarding working and commuting habits, perceived change in lifestyle habits, and mental health experiences in relation to the COVID-19 pandemic were added to the HPA in the middle/end of April. It was optional for the participants to answer the additional questions. This study included and compared data from three periods: April to June, July to September, and October to December, 2020. From the 21 April 2020 to 2 December, a total of 5599 men and women answered the additional COVID questions, and were included in the present analyses (Table 1). For comparative purposes, an additional 6232 men and women who performed a HPA during the same time period without answering the additional COVID-questions, as well as 20,864 men and women performing a HPA during the same time period in 2019 (Appendix B, Table A1), were included. The study was approved by the ethics board at the Stockholm Ethics Review Board (Dnr 2020-02727). Informed consent was obtained from the participants prior to participation.

### 2.2. Measures

The additional questions in relation to the COVID-19 pandemic are presented in Appendix A. They included questions regarding current working situation, commuting habits, and perceived change in commuting habits, as well as perceived change in sitting time, daily activity, exercise, diet, alcohol intake, and smoking due to the COVID-19 pandemic. Moreover, open questions regarding hours and minutes spent in (a) mentally passive sitting (i.e., tv-viewing, using you phone/iPad/computer to browse the internet) (b) a mentally active sitting (i.e., working, reading, solving cross-words or Sudoku), and (c) socialization (i.e., having a meal, talking with friends or family) were included, as previous studies have indicated different relationships between these different types of sedentary behavior and mental well-being [24]. Finally, questions regarding health anxiety (SHAI) [25], in terms of both one’s own health and that of relatives (modified from SHAI); employment [26] and economic [27] concerns; generalized anxiety [28]; and depression [29] were included.

From the HPA, data on BMI and estimated VO_2_max [30] were derived, as well as self-reported baseline daily PA, exercise habits, sedentary behavior, diet, alcohol abuse by AUDIT-C [31], smoking habits, overall stress, perceived health, and perceived symptoms of anxiety and depression (see Appendix B). Highest educational attainment at the time for the HPA was obtained from Statistics Sweden by linking of the participants’ personal identity numbers. Occupation was reported by the participants and coded according to the Swedish Standard Classification of Occupation [32], and further dichotomized into blue- or white-collar workers.

### 2.3. Statistical Analyses

Chi-square test (percentages) or *t*-test (mean values) results were used to compare participants with HPA + COVID data and participants with only HPA data during the study period (21 April and 2 December 2020), as well as all participants with HPA data during the study period and participants with HPA data between the same dates in 2019. Differences in working situation, commuting habits, mental health and sitting time between subgroups (Table 2) were tested using a Chi-square test (percentages), or *t*-test (mean values). Wave 1 of the COVID-19 pandemic was defined as 21st of April to 30th of June, and wave 2 as 1st of October to 2nd of December, which corresponds to the two clear wave-shapes of hospitalization due to COVID-19 in Sweden according to the Public Health Agency of Sweden [33]. From 1 July to 30 September was defined as months between the two waves, with significantly lower incidence of COVID-19. Multinomial regression modelling was used to calculate odds ratios (ORs) with 95% confidence intervals (CIs) for self-reported perceived change in six different lifestyle habits due to the COVID-19 pandemic in association to sex (women vs. men), age group (18–59 years vs. 60–78 years), educational level (University vs. non-university), occupation group (white collar vs. blue collar), baseline level of each habit, and wave of COVID-19 compared to the summer months (April–June vs. July–Septemberand October–December vs. July–September) (Table 2). Clustering of negative and positive perceived changes in lifestyle habits, respectively, were defined as negative or positive change in two or more lifestyle habits compared to less. Daily activity was not included in the clustered variable, as change in time spent sitting and daily activity are interchangeably occurring (sitting less leads to more daily activity and vice versa). Moreover, odds ratio (OR) and 95% CI was calculated using logistic regression modelling to study the association of dichotomized mental ill-health variables in relation to sex, age group, educational level, occupation group, wave of COVID-19 pandemic, type of sitting, and perceived change in lifestyle habits. The mental health variables were dichotomized to describe mental ill-health according to the following: “Frequent health anxiety, own” (Question 7A in Appendix A, answer of reply 3 or 4 vs. 1), “Frequent health anxiety, relatives” (Question 7B, reply 3 or 4 vs. 1), “Frequent anxiety symptoms” (Question 10A, reply 3 or 4 vs. 1), “Frequent depression symptoms” (Question 10B, reply 3 or 4 vs. 1), “High concerns employment” (Question 8, reply 4 or 5 vs. 1), and “High concerns economy” (Question 9, reply 4 or 5 vs. 1). Significance level was set as α < 0.05. Data were analyzed using SPSS (version 26), R 4.0.3 (R Core Team, 2020) with the Tidyverse library [34].

## 3. Results

A total of 11,831 men and women performed a HPA during the study period. Of these, 5599 (47%) answered the additional COVID-19 related questions (Table 1). There were small, albeit statistically significant, differences between individuals answering (included in the present analyses) and not answering (excluded) the extra COVID-19 related questions. Compared to nonincluded individuals, included individuals compromised more women, were older, had a higher educational level, were more often white-collar workers, had a lower BMI, exercised more, smoked less, sat more at work and experienced more stress and symptoms of anxiety and depression (Table 1). Moreover, when comparing individuals performing an HPA in year 2020 to 2019 (a “normal” year before COVID-19), we also noted some small but significant differences (Appendix B
Table A1). Participants in 2020 were more likely to be women, older, exercised more and sat less, had a higher educational level, were more often white-collar workers, had better dietary habits, smoked less, and experienced less stress compared to HPA participants in 2019.

### 3.1. Working, Commuting Situation, and Type of Sitting at Home

Almost half of the participants answering the additional COVID-19 questions reported that their occupation required that they stay at work (Table 2). The majority reported that they did not change their commuting habits due to the pandemic, whereas 10% reported that they had changed. Of those who changed, the greatest shift was from public transport to car (54%) and to active commuting (26%). Mean reported time spent in mentally active sitting was slightly higher compared to mentally passive sitting, with less time spent sitting while socializing (131, 119, and 82 min/day). Men and blue-collar workers spent more time in mentally passive sitting and less time in mentally active sitting compared to women and white-collar workers. Participants <60 years spent more time in mentally active sitting than those ≥60 years.

### 3.2. Perceived Changes in Lifestyle Habits

Most individuals stated that they had not changed their lifestyle habits due to the COVID-19 pandemic. For time spent sitting, in daily activity, and exercise, respectively, only 5%, 9%, and 10% of the participants reported a positive change, while 18%, 20%, and 20% reported a negative change. Similarly, for diet, smoking, and alcohol intake, 7%, 3%, and 8% perceived a positive change in these lifestyle habits, while 5%, 1%, and 3% perceived a negative change. Figure 1, show changes during the first and second wave. For clustering of perceived change in lifestyle habits, 13% reported a negative change in two or more lifestyle factors, whereas 8% reported a positive change in two or more lifestyle habits.

Comparing the two waves, the odds for lifestyle changes, both negative and positive, were higher during the first wave compared to the second (Figure 1 and Table 3). For example, the odds of both a perceived positive and negative change in sitting time, daily PA, and exercise were higher during the first wave compared to the second wave. Also, the odds were higher for a perceived negative change in diet and alcohol intake during the first wave compared to the second. Demographic factors were significantly associated with changes in lifestyle habits (Table 3). Women, younger participants (<60 years), participants with a university degree, white-collar workers, and those with more adverse lifestyle habits had higher odds of changing their lifestyle due to COVID-19 pandemic.

### 3.3. Mental Health Experiences

The majority of participants had low personal health anxiety, generalized anxiety and depression symptoms, as well as concerns regarding their employment and economy, with a higher proportion experiencing health anxiety for relatives (Table 4).

Six percent had clustering of two or more variables of mental ill-health (Table 5). In general, women and participants <60 years had higher odds of mental ill-health compared to men and participants ≥60 years (Table 5), while participants with a university degree and white-collar workers had significantly lower odds of having concerns regarding employment or economy (only university degree participants) compared to their counterparts. As for perceived change in lifestyle habits, the odds of mental ill-health were higher during the first wave compared to the second.

### 3.4. Type of Sitting and Change in Lifestyle Habits in Relation to Mental Ill-Health

A negative perceived change in each lifestyle habit, compared to no or positive change, was associated with higher odds for clustered mental ill-health (Figure 2). This was seen for all separate mental ill-health variables, except that it was not observed for perceived change in smoking.

More time spent in mentally passive sitting (Tertile 3; ≥120 min/day vs. Tertile 1; 0 to 90 min/day) was associated with higher odds for all variables and clustering of mental ill-health (Table 5). No similar associations were seen for more time spent in mentally active sitting or time in sitting socializing.

## 4. Discussion

In Sweden, a country with relatively few social restrictions during the pandemic, we noted small changes in the lifestyle variables overall in a large cohort of workers during the first and second wave of the COVID-19 pandemic in 2020. When changes were present, they were more pronounced during the first wave compared to the second. We also noted that the pandemic impacted some segments of the population more than others; women, individuals <60 years, those with a university degree, white-collar workers, and those with unhealthy lifestyle habits at baseline had higher odds of changing their lifestyle habits compared to their counterparts. Negative changes in lifestyle habits, as well as more time spent in mentally passive sitting at home, were associated with higher odds of mental ill-health.

### 4.1. Changes in Lifestyle Habits in Sweden Compared to Other Countries

The present results with small changes in lifestyle habits are in line with a report in May 2020 from the Swedish National Board of Public Health, where a majority reported no change compared to before the COVID-19 pandemic, (total PA 60%, diet 71%, alcohol 79%, and smoking 77%) [35]. The decrease in daily PA (26% first wave and 20% second wave) and exercise (28% and 21%) in the present sample is noticeably lower than in a large Australian study where approximately 50% reported a decrease in PA [17]. Another study exploring the number of daily steps worldwide during the first wave (March to June 2020), concluded that Swedish citizens maintained their number of daily steps to a higher degree in comparison to other countries. For example, while the maximal decrease of average step counts was 49% in Italy, Sweden had experienced a decrease of only 7% [36]. These differences might be partly explained by differences in lockdown regulations, where Sweden implemented less harsh social restrictions with no lockdown.

The increase in sitting time (26% first wave and 17% second wave) is in line with other studies [7,8,11], and may be due to similar restrictions regarding work situations in these countries. We also investigated the previously proposed difference between mentally passive and mentally active sitting behaviors on mental well-being. For example, Hallgren et al. showed that mentally active sitting was associated with a 29% lower risk for major depressive disorders after a 13-year follow-up in middle-aged men and women, while mentally passive sitting was associated with a 26% higher risk [24]. A study comparing sitting at work (presumably predominantly mentally active sitting) and in leisure time (presumably predominantly mentally passive sitting) showed weak associations of sitting at work and frequent symptoms of anxiety and depression, while more time sitting during leisure time was associated with three to four times higher OR compared to less leisure time sitting [37]. This is comparable to the results in the present study, where more self-reported time in mentally passive sitting (<120 min/day) compared to less (0 to 90 min/day) was associated with ~60% to 100% higher risk (OR) for different mental ill-health outcomes. No similar associations were found for mentally active sitting or time sitting while socializing. Although the directions of the observed associations are not clear, possible variations between different types of sitting and mental health outcomes should be considered in future studies examining the impact of pandemic restrictions, as well as in interventions targeting sitting for mental health outcomes.

For changes in sitting time, daily PA, and exercise, it was more evident that individuals with low PA levels at baseline had higher odds of a negatively perceived change in PA due to the pandemic. This is similar to a previous study by Lesser et al., which concluded that mainly inactive individuals had become less physically active during the pandemic [23]. On the contrary, a Canadian study showed that previously active adults decreased their PA, while previously inactive adults did not change their PA due to the pandemic [38]. In contrast to other studies, women in the present study had a 36–38% higher risk of decreasing their daily PA and exercise level compared to men. There were also differences between occupational groups, where white-collar workers had higher odds of increasing daily PA and exercise, while decreasing sedentary time compared to blue-collar workers. Differences in lifestyle changes due to COVID-19 in relation to occupation groups have not been addressed in previous studies.

The small changes in diet, alcohol, and smoking habits in the present study are in line with studies from other countries [10,11,12,13,15,17,19]. However, perceived changes varied between and within subgroups. For diet, individuals with healthy diets had an approximately 80% lower risk of dietary habits deteriorating compared to individuals with poor habits. Moreover, white-collar workers were more prone to changing their diet in either direction, and had approximately 90% higher probability of worsening as well as improving their diet compared to blue-collar workers. This might be due to blue-collar workers having to be at their workplaces to a higher extent, which probably contributed to fewer possibilities to change their diet behavior compared to white-collar workers, who were able to work more from home. The large differences between blue- and white-collar workers working from home or not in this study are similar to a report from Swedish statistics. The report concluded that while 56% of individuals with a university degree or equivalent reported that they did not work from home at all, the corresponding number among individuals with occupations requiring shorter education was 97% [39].

For alcohol, young individuals in this study had higher odds of both an increase and decrease in alcohol intake, with women having a lower probability of decreasing their alcohol intake. A Canadian study concluded that younger individuals and individuals with higher educational levels had higher risks of increasing their alcohol intake compared to older individuals and those with a lower education level [15]. For smoking, our results indicated that daily smokers had a 53% higher risk of increasing their smoking compared to occasional smokers, which is in line with a small Italian study [19]. However, a study of >20,000 men and women over 16 years of age found that smokers in England were more likely to report trying to quit smoking, and rates of smoking cessation were higher than before the COVID-19 pandemic [40].

There were more pronounced changes for all lifestyle habits during the first compared to the second COVID-19 wave. As recommended restrictions in Sweden were similar during both waves, possible explanations for this might be temporal effects and change saturation. This includes perceived changes in lifestyle habits during the first wave becoming the “new normal” and that people experienced more resistance to change during the second wave [41].

As healthy lifestyle habits are important in preventing noncommunicable diseases [1,42], the need to support individuals in improving or maintaining healthy lifestyle habits during the COVID-19 pandemic in order to prevent health inequalities in society and promote national public health is emphasized.

### 4.2. Changes in Mental Health in Sweden Compared to Other Countries

We found a relatively low prevalence of mental ill-health, with 4% to 6% scoring high on health anxiety regarding their own health, generalized anxiety and depression symptoms, as well as concerns regarding employment and economy. Only health anxiety for relatives was more prevalent (12%). The findings regarding health anxiety for one’s own health are similar to the report from the Swedish National Board of Public Health in May 2020, where 5% were very worried about their own health, whereas a higher frequency for health anxiety was noted for relatives (25%) [35]. This is lower in comparison to reports from the UK, where 37% experienced poor mental well-being [43]. We found that women, participants <60 years, and those with a perceived negative change in daily PA, sitting time, exercise, diet, and alcohol consumption, were more vulnerable to mental ill-health. The higher odds of mental ill-health in women and younger age groups has been reported in previous studies [21,43], as has the association between mental health and PA [7,8,9,38], alcohol consumption [17,18], diet [17], and smoking habits [17]. Interestingly in this study, the higher odds for women and younger individuals were also seen for health concerns for their relatives, which has not been reported previously.

### 4.3. Strengths and Limitations

A strength of this study is the reasonably large cohort of women and men of different ages, with a variation in educational level and occupation. The extended period of data collection (from April to December) enabled unique comparative analyses between the two waves of the COVID-19 pandemic in the total study population, as well as in subgroups. Another strength is that the study explored different components of the PA pattern, including both sitting, daily PA, and exercise, as well as different aspects of mental health (clustered mental ill-health, anxiety concern, generalized anxiety, and depression). A limitation is that the study is that it did not have data on baseline depression or anxiety. However, the analyses adjusted for self- reported general health. A limitation is the cross-sectional design, which decreased the ability to draw conclusions of causality and temporal order. Also, we examined self-reported perceived changes in lifestyle, which are not the same as within-person change based on multiple measurements. The study population consisted of individuals who accepted answering the extra covid-19-related questions, which poses a risk of selection bias. Another limitation is that data regarding lifestyle habits and changes in lifestyle habits were based on questionnaires not validated in previous work, thus risking recall bias [44]. However, questionnaires with categorical answer modes, as used in the present study, provide better validity compared to open answers for levels of PA [45].

## 5. Conclusions

Our findings suggest only small perceived changes in lifestyle habits, including time sitting, daily PA, exercise, diet, alcohol, and smoking in men and women from the Swedish working population in relation to the first two COVID-19 pandemic waves. Both negative and positive changes were more pronounced during the first wave compared to the second. Women, individuals <60 years, those with a university degree, and white-collar workers had higher odds of changing lifestyle habits compared to their counterparts. Individuals with an unhealthy lifestyle at baseline were more likely to change their lifestyle habits negatively. Thus, changes varied between sociodemographic subgroups, suggesting a clear divergence in how the pandemic waves might have impacted individuals and society. Furthermore, negative changes in lifestyle habits tended to be associated with higher levels of mental ill-health. The perceived negative changes in health-related lifestyles is a considerable public health concern, with possible implications for further increases in health inequality and mental health challenges in the light of the COVID-19 pandemic. To strengthen the resilience of both society and individuals to future viral pandemics, there is a clear need to focus on the promotion of healthy lifestyle habits, especially in socially vulnerable groups and individuals who already have an unhealthy lifestyle.

## Figures and Tables

**Figure 1 ijerph-18-03313-f001:**
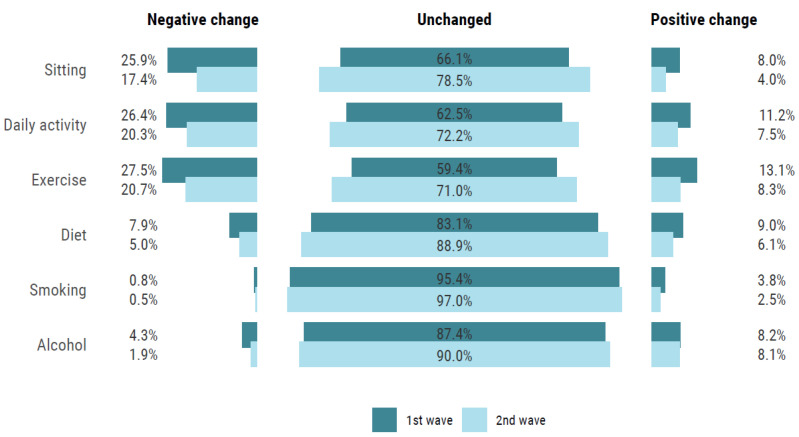
Self-reported change in lifestyle habits comparing wave 1 (April to June) and wave 2 (September to December).

**Figure 2 ijerph-18-03313-f002:**
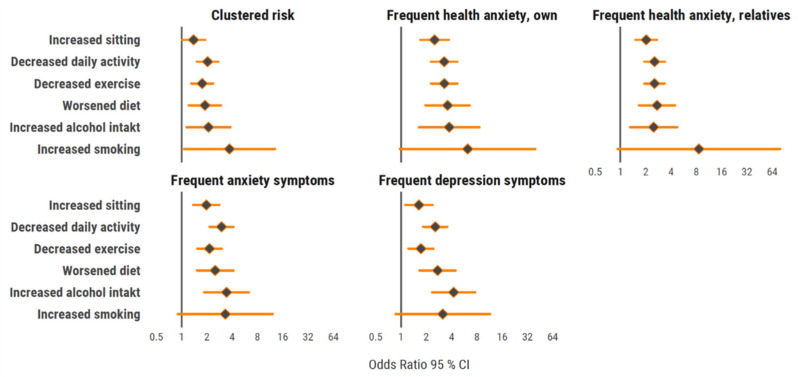
Forrest plot with odds ratio (95% CI) for clustering of mental ill-health variables in relation to change in lifestyle habits. All analyses adjusted for sex, age group, educational level, occupational group, wave of COVID-19, and baseline values for each lifestyle habit.

**Table 1 ijerph-18-03313-t001:** Participants with data from the health profile assessment (HPA) and the additional COVID-19 questions (*n* = 5599), and participants with only HPA data (*n* = 6232) between 21 April 2020 and 2 December 2021.

Title	HPA + COVID-19 Data	Only HPA Data	*p*-Value
*n*	5599	6232	
Sex (women)	50%	33%	<0.001
Age (year)	46.3 (11.0)	44.9 (11.6)	<0.001
Estimated VO_2_max (ml/min/kg)	36.0 (9.4)	35.8 (10.0)	0.518
BMI (kg/m^2^)	26.1 (4.5)	26.7 (4.8)	<0.001
Exercise habits (never/irregular)	24%	27%	<0.001
Sitting at work (all the time/75% of the time)	45%	30%	<0.001
Sitting in leisure (all the time/75% of the time)	10%	9%	0.101
University degree	35%	23%	<0.001
Occupation group (blue collar)	18%	39%	<0.001
Diet habits (very poor/poor)	4%	4%	0.060
Alcohol abuse (AUDIT-C score >4 women, >5 men)	35%	33%	0.017
Daily smoker (≥1 cig/day)	3%	7%	<0.001
Overall stress (very often/often)	13%	11%	0.001
Perceived symptoms of anxiety and depression (very often/often)	9%	7%	0.002

Data presented as mean (SD) or percentage. Differences between subgroups are tested by using Chi-square or *t*-test.

**Table 2 ijerph-18-03313-t002:** Working and commuting situation, and type of sitting at home during the study period.

	Total	Men	Women		18–59 Years	60–78 Years		White-Collar	Blue-Collar	
**Do you work from home?**										
All the time	10%	10%	10%		10%	8%		12%	1%	
Partly	26%	27%	25%		27%	20%		30%	5%	
My occupation requires that I am at work	49%	47%	52%		48%	58%		41%	90%	
I can work at home, but chose to be at work	15%	17%	13%	*p* < 0.001	15%	15%	*p* < 0.001	18%	4%	*p* < 0.001
**How have your commuting habits to and from work changed due to the COVID-19 pandemic?**										
Same as before	74%	75%	73%		74%	76%		70%	91%	
Changed	11%	9%	12%		10%	12%		12%	5%	
Stopped commuting	15%	16%	15%	*p* = 0.004	16%	12%	*p* = 0.010	18%	4%	*p* < 0.001
**If changed, how have they changed?**										
Bus/train to active commuting	26%	21%	30%		26%	29%		26%	19%	
Bus/train to car	54%	57%	52%		55%	51%		55%	57%	
Car to active commuting	8%	12%	6%		8%	11%		9%	8%	
Car to bus/train	2%	0%	3%		2%	1%		2%	0%	
Active commuting to car	8%	9%	8%		8%	7%		8%	11%	
Active commuting to bus/train	2%	1%	2%	*p* = 0.009	2%	1%	*p* = 0.930	1%	5%	*p* = 0.232
**Type of sitting at home**										
Mentally passive (min/day)	119 (78)	127 (82)	112 (73)	*p* < 0.001	119 (77)	122 (84)	*p* = 0.424	115 (74)	134 (87)	*p* < 0.001
Mentally active (min/day)	131 (174)	124 (167)	139 (179)	*p* = 0.001	134 (177)	114 (143)	*p* = 0.002	144 (182)	70 (107)	*p* < 0.001
Socializing (min/day)	82 (68)	84 (68)	81 (68)	*p* = 0.006	83 (69)	79 (62)	*p* = 0.173	81 (64)	85 (83)	*p* = 0.145

Data presented as percentage or mean (SD). Significant differences between subgroups are tested by using Chi-square test (percentages) or *t*-test (mean values).

**Table 3 ijerph-18-03313-t003:** Odds ratio (95% CI) for change in six different lifestyle habits in relation to sex, age group, educational level, occupation group, baseline level of each habits, as well as wave of COVID-19 (no change as reference).

	Negative Change in Lifestyle Habits OR (95% CI)	Positive Change in Lifestyle Habits OR (95% CI)
**Clustering of change in lifestyle habits ^§^**	Negative change in 2 or more vs. less	Positive change in 2 or more vs. less
Women vs. Men	1.25 (1.03–1.52)	1.12 (0.91–1.38)
18–59 y vs. 60–78 y	1.33 (0.97–1.83)	1.99 (1.34–2.95)
University vs. non-university	1.30 (1.07–1.58)	1.10 (0.89–1.36)
White collar vs. Blue collar	1.67 (1.21–2.30)	1.74 (1.25–2.43)
April–June vs. July–September	1.99 (1.55–2.55)	1.21 (0.94–1.56)
October–December vs. July–September	1.39 (1.11–1.75)	0.73 (0.58–0.93)
**Time spent sitting** (*n* = 4590)	Increased	Decreased
Women vs. Men	1.01 (0.86–1.19)	1.12 (0.84–1.48)
18–59 y vs. 60–78 y	1.36 (1.04–1.77)	0.92 (0.62–1.38)
University vs. non-university	1.61 (1.37–1.90)	1.17 (0.88–1.55)
White collar vs. Blue collar	1.75 (1.35–2.28)	2.44 (1.47–4.04)
Low/moderate vs. high leisure time sitting *	0.63 (0.49–0.80)	1.14 (0.69–1.89)
April–June vs. July–September	2.70 (2.20–3.32)	2.19 (1.58–3.04)
October–December vs. July–September	1.50 (1.24–1.82)	0.79 (0.56–1.10)
**Daily activity** (*n* = 4576)	Decreased	Increased
Women vs. Men	1.38 (1.17–1.61)	1.06 (0.85–1.32)
18–59 y vs. 60–78 y	0.90 (0.71–1.12)	1.48 (1.02–2.15)
University vs. non-university	1.10 (0.93–1.29)	1.05 (0.84–1.31)
White collar vs. Blue collar	1.08 (0.86–1.36)	2.03 (1.41–2.91)
Low/moderate vs. high leisure time sitting *	0.65 (0.52–0.82)	1.72 (1.11–2.68)
April–June vs. July–September	2.19 (1.80–2.68)	1.47 (1.13–1.91)
October–December vs. July–September	1.45 (1.21–1.74)	0.74 (0.58–0.95)
**Exercise** (*n* = 4591)	Decreased	Increased
Women vs. Men	1.36 (1.16–1.60)	1.03 (0.84–1.27)
18–59 y vs. 60–78 y	1.00 (0.79–1.25)	1.29 (0.91–1.83)
University vs. non-university	1.00 (0.85–1.18)	1.12( 0.91–1.38)
White collar vs. Blue collar	1.16 (0.93–1.46)	1.93 (1.36–2.74)
≥3 times/week of exercise vs. less	0.65 (0.53–0.79)	4.38 (3.07–6.23)
1–2 times/week of exercise vs. less	1.67 (1.38–2.02)	2.46 (1.67–3.64)
April–June vs. July–September	2.39 (1.95–2.92)	1.38 (1.08–1.77)
October–December vs. July–September	1.50 (1.25–1.80)	0.67 (0.53–0.85)
**Diet** (*n* = 4579)	Impaired	Improved
Women vs. Men	1.17 (0.89–1.54)	1.16 (0.91–1.48)
18–59 y vs. 60–78 y	1.39 (0.88–2.21)	1.78 (1.15–2.76)
University vs. non-university	1.27 (0.97–1.67)	1.04 (0.81–1.33)
White collar vs. Blue collar	1.93 (1.22–3.06)	1.91 (1.27–2.86)
Good vs. poor diet ^#^	0.19 (0.13–0.30)	1.12 (0.54–2.32)
April–June vs. July–September	2.02 (1.45–2.81)	1.27 (0.95–1.69)
October–December vs. July–September	1.08 (0.78–1.50)	0.71 (0.54–0.94)
**Alcohol intake** (*n* = 5171)	Decreased	Increased
Women vs. Men	0.60 (0.41–0.86)	0.90 (0.72–1.13)
18–59 y vs. 60–78 y	1.99 (1.01–3.95)	2.65 (1.68–4.20)
University vs. non-university	1.07 (0.74–1.55)	1.04 (0.83–1.30)
White collar vs. Blue collar	1.24 (0.76–2.02)	1.04 (0.77–1.41)
April–June vs. July–September	1.93 (1.27–2.92)	1.18 (0.89–1.58)
October–December vs. July–September	0.85 (0.56–1.30)	1.14 (0.89–1.44)
**Smoking** (*n* = 4505)	Decreased	Increased
Women vs. Men	1.28 (0.47–3.48)	1.42 (0.96–2.11)
18–59 y vs. 60–78 y	-	1.02 (0.58–1.81)
University vs. non-university	3.14 (1.03–9.53)	0.77 (0.50–1.19)
White collar vs. Blue collar	0.74 (0.23–2.42)	0.79 (0.48–1.28)
Never/occasionally vs. Daily smoker	0.00 (0.00–0.01)	0.23(0.12–0.44)
Occasionally smoker vs. Daily smoker	0.19 (0.07–0.53)	1.53 (0.76–3.10)
April–June vs. July–September	2.47 (0.82–7.44)	1.44 (0.91–2.29)
October–December vs. July–September	1.32 (0.40–4.35)	1.09 (0.71–1.67)

Note: All analyses mutually adjusted for sex, age group, educational level, occupational group, wave of COVID-19, and baseline values for each lifestyle habit (except for alcohol, see text and Appendix B
Table A2). * HPA question regarding sitting in leisure, coded as Low/moderate = “Almost no time”, “25% of time”, “50% of time” and High = “75% of time”, “All the time”. ^#^ HPA question regarding diet, coded as Good = “Very good” or “Good” and Poor = “Neither good or bad”, “Poor”, “Very poor”. ^§^ Including change in time spent sitting, exercise, diet, alcohol, and smoking.

**Table 4 ijerph-18-03313-t004:** Health experiences during the study period in the total population, as well as in relation to sex, age, and occupational group.

	Total	Men	Women		18–59 Years	60–78 Years		White-Collar	Blue-Collar	
**Health anxiety, own**										
I do not worry	46%	52%	41%		47%	45%		45%	52%	
I spend a lot/most of the time worrying	5%	4%	6%	*p* < 0.001	5%	3%	*p* = 0.010	5%	5%	*p* < 0.001
**Health anxiety, relatives**										
I do not worry	22%	27%	16%		21%	25%		21%	25%	
I spend a lot/most of the time worrying	12%	8%	15%	*p* < 0.001	12%	8%	*p* = 0.002	12%	10%	*p* = 0.006
**Generalized anxiety**										
Not at all	80%	85%	75%		80%	81%		80%	82%	
More than half of the days/Almost every day	4%	3%	5%	*p* < 0.001	4%	4%	*p* = 0.945	4%	3%	*p* = 0.149
**Depression symptoms**										
Not at all	73%	80%	67%		73%	78%		73%	77%	
More than half of the days/Almost every day	4%	3%	5%	*p* < 0.001	4%	3%	*p* = 0.008	4%	4%	*p* = 0.014
**Concerns employment**										
Not at all	75%	76%	74%		74%	83%		75%	71%	
Worry alot	5%	4%	5%	*p* = 0.147	5%	4%	*p* < 0.001	4%	6%	*p* = 0.019
**Concerns economy**										
Not at all	65%	66%	63%		63%	76%		65%	64%	
Worry a lot	6%	5%	7%	*p* = 0.003	7%	4%	*p* < 0.001	6%	6%	*p* = 0.856

Significant differences between subgroups are tested by using Chi-square test.

**Table 5 ijerph-18-03313-t005:** Odds ratio (95% CI) for clustering of mental-ill health, as well as each individual mental ill-health variable, in relation to sex, age, occupational group, educational level, and wave of COVID-19 pandemic (above) and time sitting and engaging in either mentally passive, mentally active, or socializing activities (below).

	Clustered Risk ≥2 vs. Less *	Frequent Health Anxiety		Anxiety Symptoms	Depression Symptoms	High Concerns Employment	High Concerns Economy
		Own	Relatives				
Women vs. Men	2.32 (1.70–3.17)	2.15 (1.50–3.07)	3.06 (2.44–3.84)	2.60 (1.87–3.63)	2.69 (1.94–3.72)	1.48 (1.11–1.97)	1.56 (1.21–2.00)
18–59 yrs vs. 60–78 yrs	1.94 (1.15–3.28)	2.17 (1.13–4.19)	1.90 (1.33–2.72)	1.12 (0.71–1.75)	1.83 (1.07–3.14)	1.50 (0.97–2.34)	1.88 (1.25–2.83)
University vs. non-university	0.82 (0.61–1.11)	1.30 (0.91–1.86)	0.87 (0.69–1.09)	0.73 (0.53–1.01)	0.89 (0.65–1.21)	0.68 (0.50–0.92)	0.64 (0.49–0.83)
White collar vs. Blue collar	0.94 (0.62–1.44)	0.67 (0.42–1.08)	0.93 (0.68–1.26)	1.05 (0.66–1.67)	0.74 (0.49–1.13)	0.69 (0.48–0.98)	0.93 (0.67–1.29)
April-June vs. July-Sept	1.49 (1.03–2.16)	2.17 (1.42–3.34)	2.87 (2.16–3.81)	1.18 (0.79–1.78)	1.63 (1.11–2.40)	0.93 (0.64–1.33)	1.36 (0.99–1.86)
October-December vs. July–September	1.39 (0.99–1.93)	1.44 (0.97–2.13)	1.32 (1.04–1.69)	1.30 (0.93–1.81)	1.34 (0.96–1.89)	0.91 (0.67–1.22)	1.17 (0.89–1.54)
Perceived good health vs. poor health	0.11 (0.08–0.14)	0.02 (0.01–0.03)					
**Time in mentally passive sitting**							
T1; 0 to 90 min/day	1.00 (ref)	1.00 (ref)	1.00 (ref)	1.00 (ref)	1.00 (ref)	1.00 (ref)	1.00 (ref)
T2; 90 to 120 min/day	0.89 (0.62–1.27)	1.36 (0.90–2.05)	1.51 (1.14–1.99)	1.05 (0.69–1.59)	0.89 (0.57–1.38)	1.24 (0.85–1.80)	1.44 (1.04–1.99)
T3; >120 min day	1.59 (1.12–2.25)	1.82 (1.19–2.80)	2.00 (1.48–2.71)	1.62 (1.07–2.46)	1.67 (1.10–2.52)	1.77 (1.21–2.58)	2.09 (1.50–2.92)
**Time in mentally active sitting**							
Tertile 1; 0 to 30 min/day	1.00 (ref)	1.00 (ref)	1.00 (ref)	1.00 (ref)	1.00 (ref)	1.00 (ref)	1.00 (ref)
Tertile 2; 30 to 90 min/day	0.98 (0.67–1.34)	1.06 (0.69–1.61)	1.09 (0.82–1.45)	1.10 (0.73–1.66)	0.88 (0.58–1.35)	0.93 (0.65–1.32)	0.83 (0.61–1.14)
Tertile 3; >90 min/day	1.15 (0.82–1.60)	1.36 (0.91–2.04)	1.27 (0.96–1.67)	1.27 (0.85–1.89)	1.15 (0.78–1.71)	1.08 (0.76–1.54)	1.03 (0.76–1.40)
**Time in sitting socializing**							
Tertile 1; 0 to 60 min/day	1.00 (ref)	1.00 (ref)	1.00 (ref)	1.00 (ref)	1.00 (ref)	1.00 (ref)	1.00 (ref)
Tertile 2; 60 to 90 min/day	0.93 (0.45–1.90)	1.13 (0.53–2.42)	0.93 (0.54–1.61)	0.72 (0.30–1.71)	0.73 (0.30–1.78)	0.70 (0.30–1.62)	0.68 (0.33–1.42)
Tertile 3; >90 min/day	1.01 (0.75–1.36)	0.91 (0.64–1.29)	1.13 (0.89–1.43)	0.81 (0.56–1.17)	0.74 (0.51–1.07)	1.17 (0.86–1.59)	0.85 (0.65–1.13)

Note: All analyses mutually adjusted for sex, age group, educational level, occupational group, and wave of COVID-19. Clustered risk and frequent personal health anxiety were additionally adjusted for baseline of perceived health. Time in mentally passive and active sitting, as well as when socializing, were additionally adjusted for baseline level of total sedentary behavior.

## Data Availability

The datasets generated and/or analyzed during the current study are not publicly available due being property of HPI Health Profile Institute, but are available from the corresponding author or the HPI Health Profile Institute on support@hpihealth.se.

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
