# Peer review of "Lifestyle Habits and Mental Health in Light of the Two COVID-19 Pandemic Waves in Sweden, 2020"

_ijerph, 2021, doi:10.3390/ijerph18063313_

Round 1
Reviewer 1 Report
This study examines changes in lifestyle habits and mental health during the pandemic. The study has several strengths, including large and demographically varied sample. However, the study also has a bit of weakness, detailed below, which should be addressed prior to publication.
Introduction
p.2 L69-71. “…However, the study also has several weaknesses, detailed below, which should be addressed prior to publication…”.
This study did not test the association between variables. Please edit the paragraph.
I suggest the author(s) carefully revise the introduction section to tighten the language and better summarize existing literature.
Methods
p.2. Study population. I would liked more information on the demographic of the sample. At present, the limited information we have is whether they are male or female. Given the number of factors which can influence changes in lifestyle habits and mental health in these unprecedented times, I would like more information on who the participants in this sample are.
Discussion
Similar to the introduction, authors should take care to revise and tighten the language used throughout the discussion to increase ease of reading.
The implication feels a little shallow and would benefit from more of practical discussion in terms of public health.
Author Response
Reviewer 1
This study examines changes in lifestyle habits and mental health during the pandemic. The study has several strengths, including large and demographically varied sample. However, the study also has a bit of weakness, detailed below, which should be addressed prior to publication.
Authors: Thanks for your time and effort invested in our work. The revised version of the manuscript underwent changes (marked in bold). Your comments and queries have been meaningful in our work to improve the manuscript.
Introduction
p.2 L69-71. “…However, the study also has several weaknesses, detailed below, which should be addressed prior to publication…”.
This study did not test the association between variables. Please edit the paragraph.
Authors: Thank you for this comment. The paragraph have now been rephrased (page 2, row 74-75).
I suggest the author(s) carefully revise the introduction section to tighten the language and better summarize existing literature.
Authors: Thank you for your comment. The introduction have now been condensed in order to better summarize the existing literature.
Methods
p.2. Study population. I would liked more information on the demographic of the sample. At present, the limited information we have is whether they are male or female. Given the number of factors which can influence changes in lifestyle habits and mental health in these unprecedented times, I would like more information on who the participants in this sample are.
Authors: We agree. Demographic information is now added in the result section, Table 1.
Discussion
Similar to the introduction, authors should take care to revise and tighten the language used throughout the discussion to increase ease of reading.
Authors: Thank you for your comment. The discussion have now been re-phrased and condensed in order to improve the ease of reading.
The implication feels a little shallow and would benefit from more of practical discussion in terms of public health.
Authors: Thank you for this comment. We have rephrased the conclusions to underscore the public health concern of the study results (page 13 row 391-397).
Reviewer 2 Report
There are many similar studies that have been published. The present study is also rather limited in scope, relies only on self-reported data and does not add anything to what we already know. I also share concerns about the study design and analysis of results.
Specific comments:
- Abbreviations such as "COVID-19" must be properly defined in the first instance of its use. Note that "COVID-19" is actually short for "Coronavirus Disease 2019".
- The general health benefits of physical activity, be it a short-bout or long-term should be briefly stated in the introduction section. For example, exercise has been linked to increased blood flow to the brain and neurotransmitter levels, enhanced plasticity and better focus, attention and information processing in typically-developing children and children with attention-deficit/hyperactivity disorder (citation: pubmed.ncbi.nlm.nih.gov/28917364).
- "perceived changes in sitting" - is there a better way to phrase this? Sedentary lifestyle?
- "Wave 1 of the COVID-19 pandemic was defined as 21st of April to 30th of June, and Wave 2 as 1st of October to 2nd of December, which corresponds to the two clear wave-shapes of hospitalization due to COVID-19 in Sweden according to the Public Health Agency of Sweden [27]" - a figure should be included to more clearly illustrate this.
- Related to the above point, any differences in restrictions/public health measures instituted during these two supposedly discrete periods should be highlighted. I am not convinced that these two periods can be taken to be disparate or significantly different.
- What were the study response and attrition rates? This was unclear.
- How do you judge "mentally active sitting" as opposed to "mentally passive sitting"?
- "Comparing the two waves, the odds for lifestyle changes, both negative and positive, were higher during the first wave compared to the second" - rather than analyse the results as two separate time periods, it should be regarded in totality as a longitudinal period. Any change is probably more pronounced during the first wave compared to the second wave because it could point towards change saturation, change fatigue etc. People are likely to be more desensitized and apathetic towards COVID-related disruptions over time.
- Did the authors adjust for baseline depression or anxiety as a covariate? Additionally, socioeconomic status still varies over time in this age range.
- The underlying data should be made publicly available. If this was not possible, please provide a reason why.
Author Response
Comments and Suggestions for Authors
There are many similar studies that have been published. The present study is also rather limited in scope, relies only on self-reported data and does not add anything to what we already know. I also share concerns about the study design and analysis of results.
Authors: Thanks for your time and effort invested in our work. We agree that there are similar studies published on this topic. However, as different governments have employed varying countermeasures and social restrictions, it is important to study the effects on lifestyle habits and health experiences in the context of different countries. Sweden has caught the attention world wide by being one of the countries implementing mainly recommended restrictions and not lock-down, and the results from such strategy is highly relevant from many aspects. Previous studies from the Swedish population is lacking, both regarding the many aspects of lifestyle and comparing the first and second wave. The revised version of the manuscript underwent changes (marked in bold). Your comments and queries have been meaningful in our work to improve the manuscript.
Specific comments:
- Abbreviations such as "COVID-19" must be properly defined in the first instance of its use. Note that "COVID-19" is actually short for "Coronavirus Disease 2019".
Authors: Thanks for noticing, this is now added (page 1, row 37).
- The general health benefits of physical activity, be it a short-bout or long-term should be briefly stated in the introduction section. For example, exercise has been linked to increased blood flow to the brain and neurotransmitter levels, enhanced plasticity and better focus, attention and information processing in typically-developing children and children with attention-deficit/hyperactivity disorder (citation: pubmed.ncbi.nlm.nih.gov/28917364).
Authors: We agree that regular physical activity have several positive health benefits for both adults and children. This is now added in the introduction (page 1 row 42-44).
- "perceived changes in sitting" - is there a better way to phrase this? Sedentary lifestyle?
Authors: Thank you for this comment. We agree that this could be better phrased. Sedentary behaviour, includes all activities that do not increase the energy expenditure above resting level. However, in this study, we aimed to be more specific focusing on time spend in sitting. Therefore, the definition ”time spent in sitting” will be used and has been changed throughout the manuscript.
- "Wave 1 of the COVID-19 pandemic was defined as 21st of April to 30th of June, and Wave 2 as 1st of October to 2nd of December, which corresponds to the two clear wave-shapes of hospitalization due to COVID-19 in Sweden according to the Public Health Agency of Sweden [27]" - a figure should be included to more clearly illustrate this.
Authors: Thank you for your comment. A figure could clarify the two waves, however we do not have permission from the Public Health Agency of Sweden to publish their figures and hence only make reference to it.
- Related to the above point, any differences in restrictions/public health measures instituted during these two supposedly discrete periods should be highlighted. I am not convinced that these two periods can be taken to be disparate or significantly different.
Authors: Thank you for lifting an important issue. In Sweden there were no national differences in social restrictions in between the first and second wave. This is now further highlighted in the discussion (page 11, row 336-341).
- What were the study response and attrition rates? This was unclear.
Authors: Thank you for this comment. All this information was present in Appendix table 1. However, we have now added this table in the main text as “Table 1”.
- How do you judge "mentally active sitting" as opposed to "mentally passive sitting"?
Authors: Thank you for lifting an important question. Our definitions are “Mentally passive sitting, including tv-viewing, using you phone/ipad/computer to browse the internet etc” and “Mentally active sitting, including working, reading, solving cross-words or sudoku etc”. This is described in “Appendices A” and in the methods section (page 3 row 105-107).
- "Comparing the two waves, the odds for lifestyle changes, both negative and positive, were higher during the first wave compared to the second" - rather than analyse the results as two separate time periods, it should be regarded in totality as a longitudinal period. Any change is probably more pronounced during the first wave compared to the second wave because it could point towards change saturation, change fatigue etc. People are likely to be more desensitized and apathetic towards COVID-related disruptions over time.
Authors: Thank you for lifting an important point. We agree that it is important to explore changes in a longitudinal period. Information of perceived changes in lifestyle habits over time is now added in the results (page 5, row 192-197). Like you mention, that the results are more pronounced during the first wave compared to the second may be due to other factors such as change fatigue is now further discussed in the discussion (page 11, row 336-341).
- Did the authors adjust for baseline depression or anxiety as a covariate? Additionally, socioeconomic status still varies over time in this age range.
Authors: Thank you. We agree that this would have been desirable. But unfortunately we did not have data on baseline depression or anxiety. However, we did adjust for self- reported general health as a proxy as well as for socioeconomic status. This is now added in the strengths and limitations section (page 12, row 370-371) . We agree that sociodemographic status indeed can vary but is supposed to be rather marginal in this short perspective of a year.
- The underlying data should be made publicly available. If this was not possible, please provide a reason why.
Authors: We agree of the importance of making data publicly available. This dataset generated and analyzed during the current study are not publicly available due being property of HPI Health Profile Institute, but are available from the corresponding author or the HPI Health Profile Institute on support@hpihealth.se.” This is stated in the “Data Availability Statement” (page 13, row 415-417).
Reviewer 3 Report
I thank the Editor and authors for the opportunity to review a manuscript.
This study investigates perceived changes of lifestyle habits in response to the COVID-19 pandemic and associations between perceived lifestyle changes and mental health. The cross-sectional research is based on the large cohort of women and men of different ages with a variation in educational level and occupation. Data are unique because Sweden is a country with relatively few social restrictions during the pandemic. The study includes data on study population as well as compares sub-groups. The undoubted advantage is the fact that data were collected at three time points: April to June, July to September and October to December, 2020. The results have important clinical implication to support healthy lifestyle habits of those who belong to vulnerable groups and prevent health inequalities in society.
The paper has overall a good technical content and it’s easily readable. Abstract is an informative and balanced summary of what was done and what was found. In the body of manuscript, the authors provided background that puts the manuscript into context and allows readers to understand the purpose and significance of the study and also defined the problem addressed. The manuscript is characterized by appropriate study design, statistical analysis as well as quality of results presentation, discussion and conclusions. Limitations were appropriately acknowledged. I congratulate the authors on a very interesting manuscript. I believe the importance of this paper. Well done!
Author Response
I thank the Editor and authors for the opportunity to review a manuscript.
This study investigates perceived changes of lifestyle habits in response to the COVID-19 pandemic and associations between perceived lifestyle changes and mental health. The cross-sectional research is based on the large cohort of women and men of different ages with a variation in educational level and occupation. Data are unique because Sweden is a country with relatively few social restrictions during the pandemic. The study includes data on study population as well as compares sub-groups. The undoubted advantage is the fact that data were collected at three time points: April to June, July to September and October to December, 2020. The results have important clinical implication to support healthy lifestyle habits of those who belong to vulnerable groups and prevent health inequalities in society.
The paper has overall a good technical content and it’s easily readable. Abstract is an informative and balanced summary of what was done and what was found. In the body of manuscript, the authors provided background that puts the manuscript into context and allows readers to understand the purpose and significance of the study and also defined the problem addressed. The manuscript is characterized by appropriate study design, statistical analysis as well as quality of results presentation, discussion and conclusions. Limitations were appropriately acknowledged. I congratulate the authors on a very interesting manuscript. I believe the importance of this paper. Well done!
Authors: Thanks for your time and effort invested in our work.
Round 2
Reviewer 2 Report
Thank you for the revisions.
This manuscript is a resubmission of an earlier submission. The following is a list of the peer review reports and author responses from that submission.